# LATENT DIFFUSION TRANSFORMER WITH LOCAL NEURAL FIELD AS PDE SURROGATE MODEL

**Louis Serrano[1], Jean-Noël Vittaut[2], Patrick Gallinari[1, 3]**
[1] Sorbonne Université, CNRS, ISIR, 75005 Paris, France
[2] Sorbonne Université, CNRS, LIP6, 75005 Paris, France
[3] Criteo AI Lab, Paris, France
`louis.serrano@isir.upmc.fr`
`jean-noel.vittaut@lip6.fr, patrick.gallinari@isir.upmc.fr`

## ABSTRACT

We introduce a diffusion transformer architecture, AROMA (Attentive Reduced Order Model with Attention), for dynamics modeling of complex systems. By employing a discretization-free encoder and a local neural field decoder, we construct a latent space that accurately captures spatiality without requiring traditional space discretization. The diffusion transformer models the dynamics in this latent space conditioned on the previous state. It refines the predictions providing enhanced stability compared to traditional transformers and then enabling longer rollouts. AROMA demonstrates superior performance over existing neural field methods in simulating 1D and 2D equations, highlighting the effectiveness of our approach in capturing complex dynamical behaviors.

## 1 INTRODUCTION

In recent years, many deep learning (DL) surrogate models have been introduced to approximate solutions to partial differential equations (PDEs) (Lu et al., 2021; Li et al., 2021; Brandstetter et al., 2022; Stachenfeld et al., 2022). Amongst them, the family of neural operators and operator networks has been extensively adopted and tested in various scientific domains, showing the potential of data-centric DL models in science (Pathak et al., 2022; Vinuesa & Brunton, 2022).

In essence, neural operators adopt the functional formulation of a PDE problem, yet they are implemented with architectures that exhibit constraints with respect to the discretization of the data and the geometry of the domain. To alleviate these issues, neural-field based methods have been proposed to model the dynamics of a system that follows PDEs (Yin et al., 2022; Chen et al., 2022) or more generally to learn operators between functions (Serrano et al., 2023). These models are discretization free, they can be applied on regular and irregular grids, with different samplings. However, they still face some challenges in applicability: (i) they work with auto-decoding which can be unstable during training and computationally intensive at inference, (ii) they unroll the dynamics in a latent-space over a trajectory of global codes that ignore the spatial information of the data.

In this paper, we propose a discretization-free model that offers a solution to both of these issues by taking a different approach. Our hypothesis is that considering spatiality is essential to modeling spatiotemporal phenomena, and therefore the latent space should conserve a spatial structure. For instance, state-of-the-art latent U-Net (Rombach et al., 2022) and transformer (Peebles & Xie, 2023) diffusion models operate respectively on latent images or sequences of smaller dimensions that keep a spatial property. However, in the literature these latent representations are mostly obtained with traditional convolutional neural networks (CNNs) (He et al., 2016) which are not compatible with our discretization-free constraint. For the same reason, patching observations Lee et al. (2023) is not aligned with our objective. Therefore, there is no existing architecture capable of encoding without discretization constraints function values into a latent representation that conserves a spatial dimension.

Our work presents:

- A novel architecture for latent-space modeling of dynamical systems, combining local neural fields with a diffusion transformer for enhanced stability.

- A discretization-free encoder that captures spatially-aware tokens from data, bypassing traditional patching, and a decoder that uses cross-attention with local neural fields for accurate, continuous data reconstruction.

- An innovative method for modeling dynamics in latent space, utilizing a diffusion transformer for iterative refinement.

- Detailed validation of our model's superior performance over existing frameworks, evidenced by results on two datasets and supported by a perturbation analysis showcasing the spatial significance of our latent tokens.

## 2 PROBLEM SETTING

In this paper, we focus on time-dependent PDEs defined over a spatial domain $\Omega$ (with boundary $\partial\Omega$) and temporal domain $[0, T]$. In the general form, their solutions $\boldsymbol{u}(x, t)$ satisfy the following constraints :

$$\frac{\partial \boldsymbol{u}}{\partial t} = F\left(\nu, t, x, \boldsymbol{u}, \frac{\partial \boldsymbol{u}}{\partial x}, \frac{\partial^2 \boldsymbol{u}}{\partial x^2}, \dots\right), \quad \forall x \in \Omega, \forall t \in (0, T] \tag{1}$$

$$\mathcal{B}(\boldsymbol{u})(t, x) = 0 \quad \forall x \in \partial\Omega, \forall t \in (0, T] \tag{2}$$

$$\boldsymbol{u}(0, x) = \boldsymbol{u}^0 \quad \forall x \in \Omega \tag{3}$$

Where $\nu$ represents a set of PDE coefficients, Equations (2) and (3) represent the constraints with respect to the boundary and initial conditions. We aim to learn from solutions data obtained with classical solvers the evolution operator $\mathcal{G}$ that predicts the state of the system at the next time step: $\boldsymbol{u}^{t+\Delta t} = \mathcal{G}(\boldsymbol{u}^t)$. We have access to training trajectories obtained with different initial conditions, and we want to generate accurate trajectory rollouts for new initial conditions at test time. A rollout is obtained by the iterative application of the evolution operator $\boldsymbol{u}^{m\Delta t} = \mathcal{G}^m(\boldsymbol{u}^0)$.

## 3 MODEL DESCRIPTION

Our model comprises key components to enhance the latent-space modeling of dynamical systems.

- **Encoder** : $u_{\mathcal{X}}^t \to \boldsymbol{Z}^t$. The encoder takes input values over a grid $\mathcal{X}$ and reduces the dimension to a set of $M$ multivariate normal statistics $\mathcal{E}_w(u_{\mathcal{X}}^t) = (\mu_j^t, \sigma_j^t)_{j=1}^M$ from which latent tokens can be sampled $\boldsymbol{z}_j^t \sim \mathcal{N}(\mu_j^t, (\sigma_j^t)^2)$. We note the mean, standard deviation and latent tokens $\boldsymbol{\mu}^t = (\mu_1^t, \cdots, \mu_M^t), \boldsymbol{\sigma}^t = (\sigma_1^t, \cdots, \sigma_M^t), \boldsymbol{Z}^t = (\boldsymbol{z}_1^t, \cdots, \boldsymbol{z}_M^t)$. Each token is $d$-dimensional.

- **Latent refiner** : $\boldsymbol{Z}^t \to \hat{\boldsymbol{Z}}^{t+\Delta t}$. We model the dynamics in the latent space through a latent conditional diffusion model $r_\theta$. The diffusion model $r_\theta$ is used to predict the next set of latent tokens by denoising them in $K$ steps. Starting from pure gaussian noise $\tilde{\boldsymbol{Z}}_K^{t+\Delta t} \sim \mathcal{N}(0, I)$, we reverse the diffusion process: $\tilde{\boldsymbol{Z}}_{k-1}^{t+\Delta t} = \text{DENOISE}(\tilde{\boldsymbol{Z}}_k^{t+\Delta t}, r_\theta(\boldsymbol{Z}^t, \tilde{\boldsymbol{Z}}_k^{t+\Delta t}, k))$ for $1 \le k \le K$ and set the prediction as $\hat{\boldsymbol{Z}}^{t+\Delta t} = \tilde{\boldsymbol{Z}}_0^{t+\Delta t}$. We note $\mathcal{R}_\theta$ this denoising operation. The diffusion model enables a "refinement process" throughout the denoising, improving the prediction at each denoising step. This is inspired by Lippe et al. (2023), as we experimentally observed that a probabilistic model was more robust than a deterministic one for temporal extrapolation.

- **Decoder** : $\hat{\boldsymbol{Z}}^{t+\Delta t} \to \hat{\boldsymbol{u}}^{t+\Delta t}$. The decoder uses the latent tokens $\hat{\boldsymbol{Z}}^{t+\Delta t}$ to approximate the function value $\hat{u}^{t+\Delta t}(x) = \mathcal{D}_\psi(x, \hat{\boldsymbol{Z}}^{t+\Delta t})$ for each query coordinate $x \in \Omega$. We therefore note $\hat{\boldsymbol{u}}^{t+\Delta t} = \mathcal{D}_\psi(\boldsymbol{Z}^{t+\Delta t})$ the predicted function, and $\hat{u}_{\mathcal{X}}^{t+\Delta t} = \mathcal{D}_\psi(\boldsymbol{Z}^{t+\Delta t}, \mathcal{X})$ the predicted values over the grid $\mathcal{X}$.

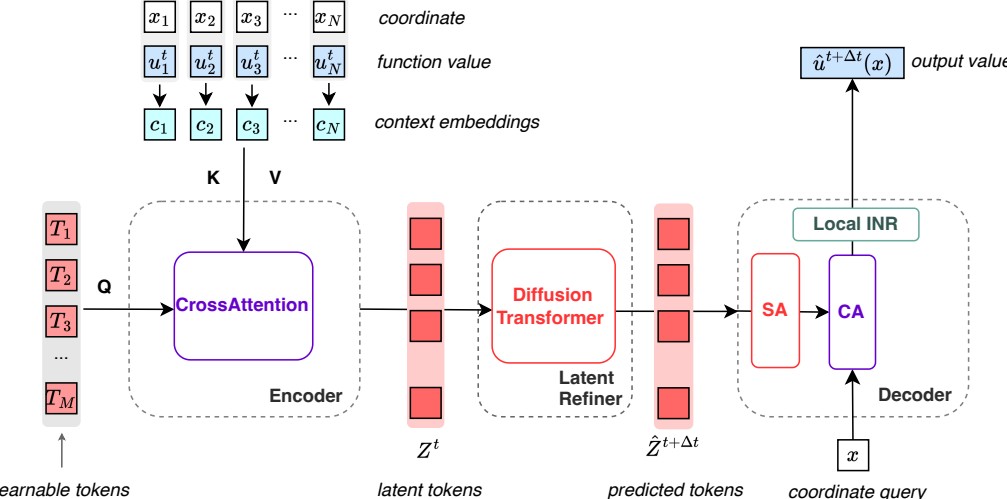

Figure 1: **AROMA inference**: The discretization-free encoder compresses the information of a set of $N$ input values to a sequence of $M$ latent tokens, where $M < N$. The conditional diffusion transformer is used to model the dynamics, acting as a latent refiner. The continuous decoder leverages self-attentions (SA), cross-attention (CA) and a local INR to get back to the physical space.

**Inference**  We encode the initial condition and unroll the dynamics in the latent space by successive denoisings: $\hat{\boldsymbol{u}}^{m\Delta t} = \mathcal{D}_\psi \circ \mathcal{R}_\theta^m \circ \mathcal{E}_w(\boldsymbol{u}^0)$. We then decode along the trajectory to get the reconstructions. We outline the full inference pipeline in Figure 1

**Training**  We do a two-stage training: first train the encoder and decoder, second train the refiner.

### 3.1 ENCODER-DECODER WITH LOCAL IMPLICIT NEURAL REPRESENTATION

**Encoder**  The encoder architecture efficiently processes input functions irrespective of their discretization (Figure 5). It shares some similarities with the Perceiver architecture (Jaegle et al., 2021) but is uniquely designed for our setting. **(i) context embedding creation**: Given an input sequence of coordinate-value pairs $(x_1, \boldsymbol{u}(x_1)), \ldots, (x_N, \boldsymbol{u}(x_N))$, we construct high-dimensional context embeddings. Each embedding $c_l$ is obtained as $c_l = f_u(\boldsymbol{u}(x_l)) + f_x(x_l)$ , where $f_u$ and $f_x$ are feedforward neural networks. **(ii) information aggregation**: The sequence of context embeddings $\boldsymbol{c} = (c_1, \ldots, c_N)$ is aggregated onto a smaller set of learnable query tokens $\mathbf{T} = (T_1, \ldots, T_M)$. This is achieved using a multihead cross-attention layer and a feedforward (FF) network, expressed as $\mathbf{T}' = \mathrm{FF}(\mathrm{CrossAttention}(\mathbf{Q} = \mathbf{W}_Q\mathbf{T}, \mathbf{K} = \mathbf{W}_K\boldsymbol{c}, \mathbf{V} = \mathbf{W}_V\boldsymbol{c}))$. **(iii) channel dimension compression**: Finally, the information in the channel dimension of $\mathbf{T}'$ is compressed using a bottleneck linear layer, yielding the mean and standard deviation tokens $\boldsymbol{\mu} = \mathbf{W}_\mu\mathbf{T}'$ and $\boldsymbol{\sigma} = \mathbf{W}_\sigma\mathbf{T}'$.

**Decoder**  The decoder architecture proceeds as follows (Figure 5). **(i) latent tokens processing**: The decoder first lifts the latent tokens $\boldsymbol{Z}$ - sampled from a multivariate gaussian with parameters $\boldsymbol{\mu}$ and $\boldsymbol{\sigma}$ - to a higher channel dimension. **(ii) self-attention layers**: Several layers of self-attention are then applied to get processed tokens $\boldsymbol{Z}'$. **(iii) cross-attention**: The decoder applies cross-attention to obtain a modulation vector that depends on the query coordinate $\phi_u(x) = \mathrm{CrossAttention}(\mathbf{Q} = \mathbf{W}_Q\gamma_q(x), \mathbf{K} = \mathbf{W}_K\boldsymbol{Z}', \mathbf{V} = \mathbf{W}_V\boldsymbol{Z}')$, where $\gamma_q$ is a feed forward neural network with fourier features of bandwidth $\omega_q$. **(iv) local implicit neural representation (INR) decoding**: Finally, an INR is used to decode the local modulation vector to obtain the output of the neural field. As in Lee et al. (2023) this INR employs a two-level decoding with two frequency bandwidths $\omega_1, \omega_2$ such that $\omega_q \leq \omega_2 \leq \omega_1$. We refer the reader to additional details regarding its implementation in Appendix B. In contrast with existing neural field methods for dynamics modeling, the modulation vector here is local, i.e. it does not modify the weights of the INR for the whole coordinate space.

**Training** The encoder and decoder are jointly optimized as a variational autoencoder (VAE) (Kingma & Welling, 2013) to minimize the following objective : $\mathcal{L} = \mathcal{L}_{\text{recon}} + \beta \cdot \mathcal{L}_{KL}$; where $\mathcal{L}_{\text{recon}} = \text{MSE}(u^t_{\mathcal{X}}, \hat{u}^t_{\mathcal{X}})$ is the reconstruction loss between the input and $D_\psi(\boldsymbol{Z}^t, \mathcal{X})$, with $\boldsymbol{Z}^t \sim \mathcal{N}(\boldsymbol{\mu^t}, (\boldsymbol{\sigma}^t)^2)$ and $\boldsymbol{\mu^t}, \boldsymbol{\sigma}^t = \mathcal{E}_w(u^t_{\mathcal{X}})$. The KL loss $\mathcal{L}_{\text{KL}} = D_{\text{KL}}(\mathcal{N}(\boldsymbol{\mu^t}, (\boldsymbol{\sigma}^t)^2) \,||\, \mathcal{N}(0, I))$ helps regularizing the network and prevents overfitting.

## 3.2 TRANSFORMER-BASED REFINER

As the latent tokens $\boldsymbol{Z}^t$ have a sequential structure, we use a conditional diffusion transformer architecture close to Peebles & Xie (2023) for $r_\theta$. At diffusion step $k$, the input to the network is a sequence stacking the tokens and the current noisy target estimate $(\boldsymbol{Z}^t, \tilde{\boldsymbol{Z}}^{t+\Delta t}_k)$. See Appendix B, Figure 3 and Figure 2 for more details. To train the refiner $r_\theta$, we freeze the encoder and decoder, and use the encoder to sample pairs of successive latent tokens $(\boldsymbol{Z}^t, \boldsymbol{Z}^{t+\Delta t})$. We employ the "v-predict" formulation of DDPM (Salimans & Ho (2022)) for training and sampling.

## 4 EXPERIMENTS

**Datasets** To evaluate our framework, we utilized two fluid dynamics datasets with unique initial conditions for each trajectory: • **1D Burgers' Equation** (*Burgers*): Models shock waves, using a dataset with periodic initial conditions and forcing term. It includes 2048 training and 128 test trajectories, at resolutions of $(250, 100)$. Training involves 10 sub-trajectories of 25 timestamps each, using only the first 20 for training and all 25 for evaluation on test trajectories. • **2D Navier-Stokes Equation** (*Navier-Stokes*): Focuses on viscous, incompressible fluid flow, with training and test sets of 256 and 16 trajectories of 40 timestamps. The original $256 \times 256$ resolution data is subsampled to $64 \times 64$, using the first 20 timestamps for training and all 40 timestamps for evaluation.

**Baselines** We compare our model to FNO (Li et al., 2021), and choose DINo (Yin et al., 2022) and CORAL (Serrano et al., 2023) as the neural field baselines. We also include a deterministic AROMA without diffusion as an ablaton study.

**Training and evaluation** During training, we use only the data fromn the training horizon (*In-t*). At test time, we evaluate the models to unroll the dynamics for new initial conditions in the training horizon (*In-t*) and for temporal extrapolation (*Out-t*).

**Results** Table 1 showcases the performance of our model against various models on both datasets. The results are evaluated using the mean squared error (MSE) metric. Notably, AROMA demonstrates superior performance for *Navier-Stokes* dataset, achieving the lowest MSE scores in both *In-t* and *Out-t* scenarios. It also shows competitive results on *Burgers*, outperforming other INR-based models and closely following FNO. AROMA without diffusion obtains good results in the training horizon but is particularly less robust on longer rollouts (Figure 7, Figure 8, Figure 9).

Table 1: **Dynamics modeling** - Test results. Metrics in MSE. Lower the better.

| dataset → | Navier-Stokes | | Burgers | |
|---|---|---|---|---|
| model ↓ | In-t | Out-t | In-t | Out-t |
| FNO | 5.68e-4 | 8.95e-3 | **3.76e-4** | **1.08e-3** |
| DINo | 1.27e-3 | 1.11e-2 | 5.82e-2 | 6.50e-2 |
| CORAL | 1.86e-4 | **1.02e-3** | 8.24e-4 | 2.56e-3 |
| AROMA | **6.95e-5** | **1.02e-3** | 3.96e-4 | 1.29e-3 |
| AROMA no refiner | 8.21e-5 | 1.70e-3 | 5.79e-4 | 2.04e-3 |

## 5 CONCLUSION

AROMA offers a novel approach to neural field modeling, incorporating attention mechanisms and a latent diffusion transformer for spatio-temporal dynamics. It has shown improved performance on small-size datasets and holds potential for effective scaling to larger datasets, thanks to its use of transformer blocks.

ACKNOWLEDGEMENTS

We acknowledge the financial support provided by DL4CLIM (ANR-19-CHIA-0018-01), DEEP-NUM (ANR-21-CE23-0017-02), PHLUSIM (ANR-23-CE23-0025-02), and PEPR Sharp (ANR-23-PEIA-0008, ANR, FRANCE 2030).

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

## A    RELATED WORK

Our model differs from existing models in the field of operator learning and more broadly from existing neural field architectures. The works most related to ours are the following.

**Transformer for PDE**    Hao et al. (2023) introduced a versatile transformer architecture specifically tailored for operator learning. Their design incorporates an attention mechanism and employs a mixture of experts strategy, effective in addressing multi-scale challenges. However, their architecture resorts to linear attention as they do not reduce spatial dimensions, and each attention layer has a linear complexity in the sequence size, which can be prohibitive for very deep networks.

**Local Neural Fields**    We are not the first work that proposes to leverage locality to improve the design of neural fields. In a different approach, Bauer et al. (2023) proposed a grid-based latent space where the modulation function $\phi$ is dependent on the query coordinate $x$. This concept enables the application of architectures with spatial inductive biases for generation on the latent representations, such as a U-Net Denoiser for diffusion processes. Similarly, Lee et al. (2023) developed a locality-aware, generalizable Implicit Neural Representation (INR) with demonstrated capabilities in generative modeling. Both of these architectures assume regular input structures, be it through patching methods or grid-based layouts. Expanding on this theme, Zhang et al. (2023) explored an analogous concept but diverged by implementing an aggregating process inspired from the Perceiver architecture Jaegle et al. (2021). Their approach also leverages cross-attention instead of patching but their architecture differs from ours as it is designed for processing point-cloud shapes while our blocks can be used with arbitrary functions.

## B    ARCHITECTURE DETAILS

**Diffusion transformer**    We illustrate how our diffusion transformer is trained and used at inference in Figure 2 and Figure 3. We provide the diffusion step $k$ which acts as a conditioning input for the diffusion model. We use an exponential decrease for the noise level as in Lippe et al. (2023) i.e. $\alpha_k = 1 - \sigma_{\min}^{k/K}$.

**Encoder-Decoder**    We provide a more detailed description of the encoder-decoder pipeline in Figure 5.

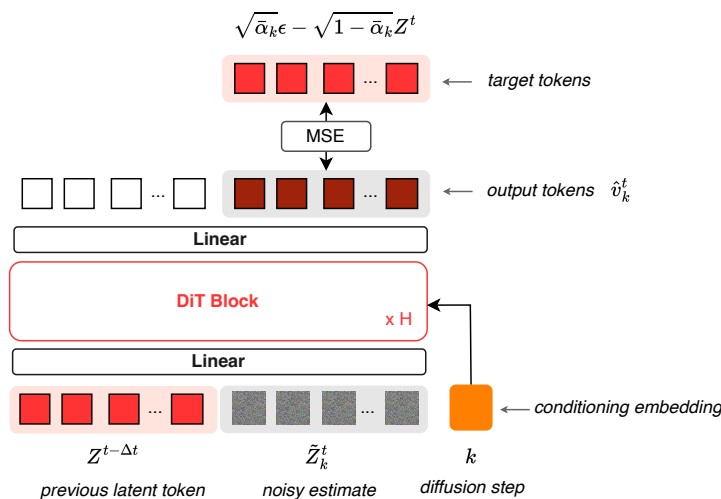

Figure 2: During training, we noise the next-step latent tokens $\mathbf{Z}^{t+\Delta t}$ and train the transformer to predict the "velocity" of the noise. Each DIT block is implemented as in Peebles & Xie (2023).

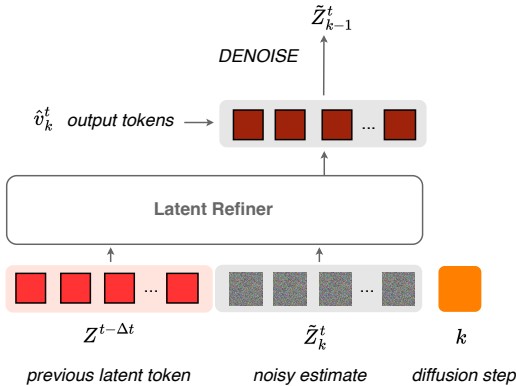

Figure 3: At inference, we start from $\tilde{\mathbf{Z}}_K^{t+\Delta t} \sim \mathcal{N}(0, I)$ and reverse the diffusion process to denoise our prediction. We set our prediction $\hat{\mathbf{Z}}^{t+\Delta t} = \tilde{\mathbf{Z}}_0^{t+\Delta t}$.

**Local INR**   We show the implementation of the local INR from Lee et al. (2023) in Figure 4.

## C   DATASET DETAILS

**1D Burgers' Equation**   (*Burgers*): This equation models shock waves without diffusion, expressed as $\frac{\partial u}{\partial t} + 0.5 \frac{\partial u^2}{\partial x} = f$. We take the same dataset as in Brandstetter et al. (2022) to guarantee periodicity of the initial conditions and forcing term $f$. Our training and test sets consist of 2048 and 128 trajectories, respectively, reduced to temporal and spatial resolutions of $(250, 100)$. For training, we create 10 sub-trajectories of size 25, from which we keep only the first 20 timestamps. For evaluation we use the 25 timestamps.

**2D Navier-Stokes Equation**   (*Navier-Stokes*): This dataset (Yin et al., 2022) deals with the flow of viscous, incompressible fluid and is represented by the equation $\frac{\partial w}{\partial t} + u \cdot \nabla w = \nu \Delta w + f$, $\nabla u = 0$ for $x \in \Omega, t > 0$, where $\omega$ is the vorticity, $\nu = 10^{-3}$ is the viscosity and $f$ is a forcing term. The training and testing datasets have 256 and 16 trajectories. We keep the first 20 timestamps for

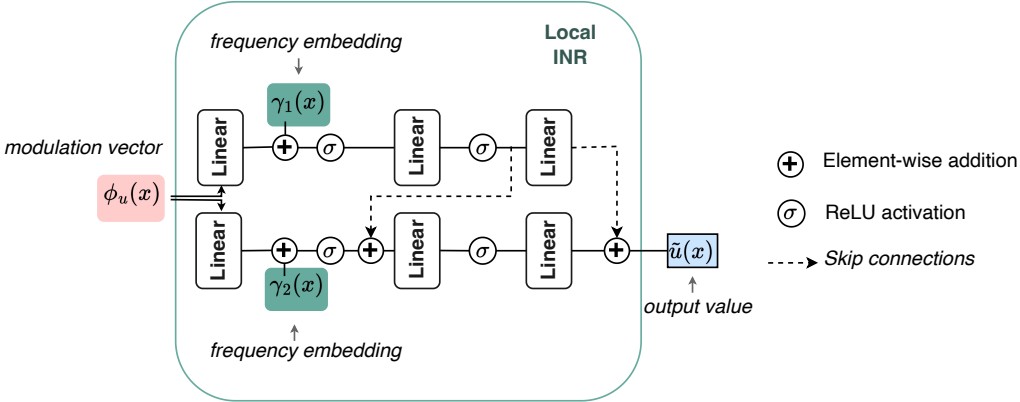

Figure 4: Architecture of the local INR. $\gamma_1$ and $\gamma_2$ are fourier embeddings of the coordinate $x$ with bandwidths $\omega_2 \leq \omega_1$. The output of the INR is residual as we use skip connections.

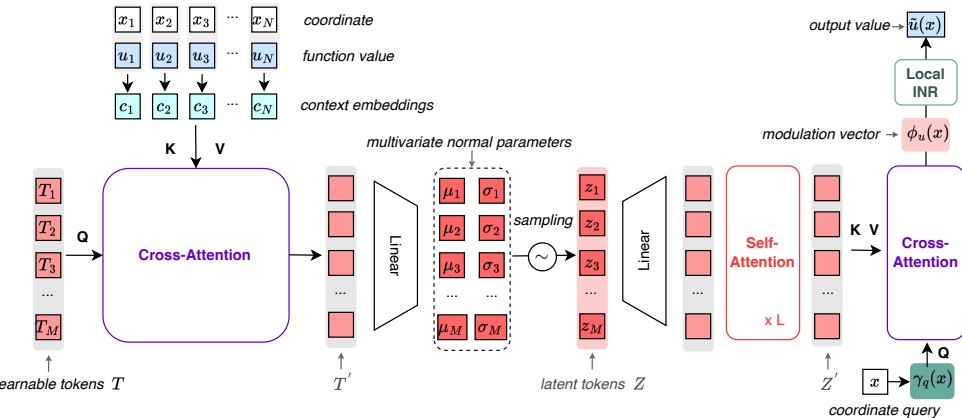

Figure 5: Architecture of our encoder and decoder. We regularize the architecture as a variational auto-encoder. Cross-attention layers are used to aggregate the $N$ observations into $M$ latent tokens, and to expand the $M$ processed tokens to the queried values. We use a bottleneck layer to reduce the channel dimension of the latent space.

training and evaluate over 40 timestamps. The original data is at a $256 \times 256$ resolution, which we sub-sample to achieve a frame size of $64 \times 64$.

## D    ADDITIONAL RESULTS

### D.1    LONG ROLLOUTS

On *Burgers*, we test the capacity of the models to unroll the dynamics from the 50th timestamp for 200 steps up to timestamp 250. Note that none of the models have been trained for this task, and that the models have not seen 20% of the data within this horizon. As showcased in Table 1 the performance on this dataset quickly deteriorates in extrapolation as the intensity and location of the shocks are hard to predict, and therefore the performance of all models is relatively poor. Still, AROMA performs the best with a test MSE over the whole trajectory of 0.119 compared to 0.142 for FNO. Most notably this analysis helps in motivating the modeling choice of our diffusion transformer. While the predictions with FNO and AROMA are relatively close to each other in Figure 6 and Figure 7, AROMA without diffusion quickly diverges from the ground truth (Figure 8) highlighting the increased stability with the refinement process. We show in Figure 9 that AROMA's

and FNO's predictions are still highly correlated at the end of the trajectory while AROMA without diffusion cannot cope with long rollouts.

For *Navier-Stokes*, we show an example of test trajectory in the training horizon (Figure 10) and in extrapolation (Figure 11).

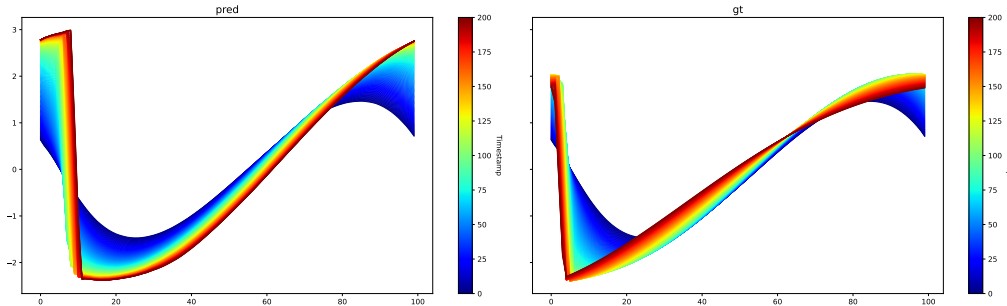

Figure 6: Test example long rollout trajectory with AROMA on *Burgers*. Left is the predicted trajectory and right is the ground truth.

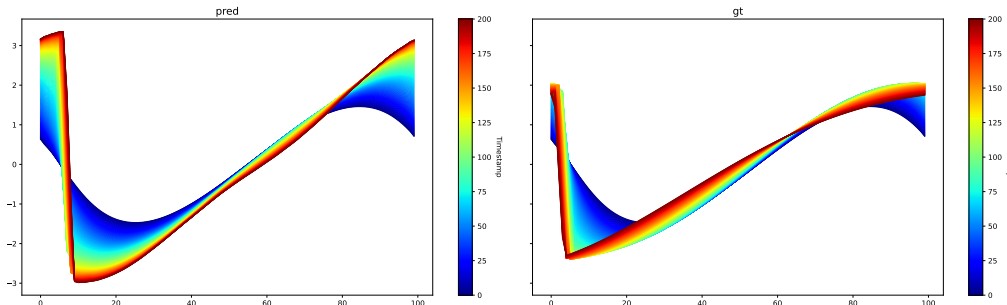

Figure 7: Test example long rollout trajectory with FNO on *Burgers*. Left is the predicted trajectory and right is the ground truth.

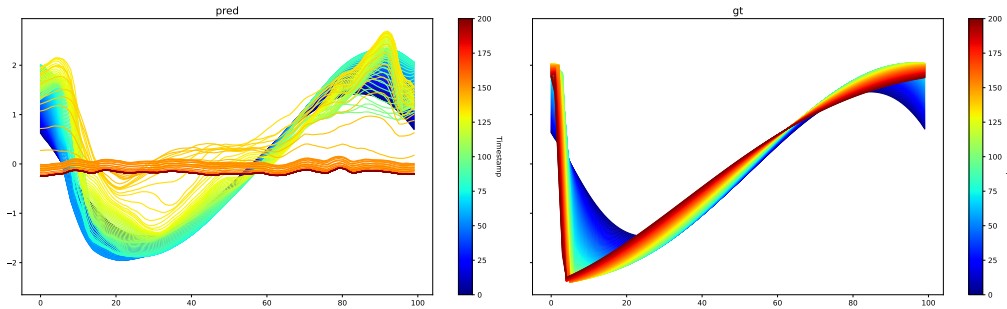

Figure 8: Test example long rollout trajectory with AROMA without diffusion on *Burgers*. Left is the predicted trajectory and right is the ground truth.

## D.2 PERTURBATION ANALYSIS

To validate the spatial interpretation of our latent tokens, we establish a baseline code $\boldsymbol{Z}^0$, and introduce perturbations by sequentially replacing the $j$-th token, $\boldsymbol{z}_j^0$, with subsequent tokens along the trajectory, denoted as $\boldsymbol{z}_j^1, \boldsymbol{z}_j^2, \ldots, \boldsymbol{z}_j^t$. Thus, the perturbed tokens mirrors $\boldsymbol{Z}^0$ in all aspects except for the $j$-th token, which evolves according to the true token dynamics. We show reconstruction visualizations of the perturbed tokens in Figure 12, Figure 13, Figure 14, Figure 15, Figure 16,

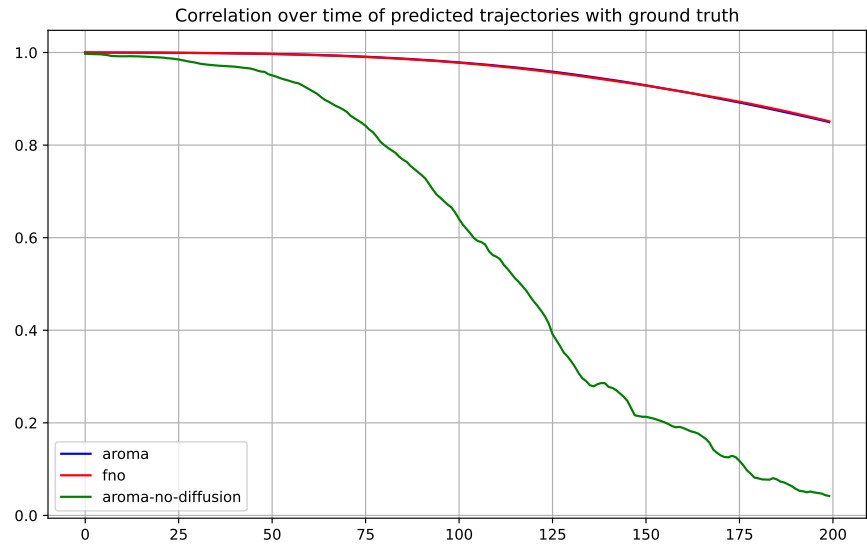

Figure 9: Correlation of predicted trajectories with ground truth.

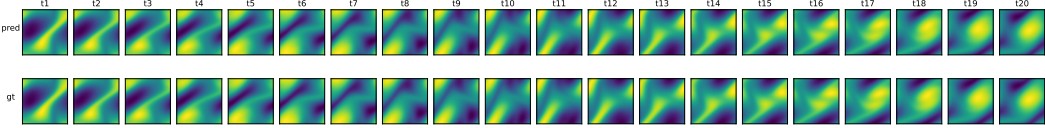

Figure 10: Test example rollout trajectory with AROMA on *Navier-Stokes* and *In-t*. The first row is the prediction and the second row is the ground truth.

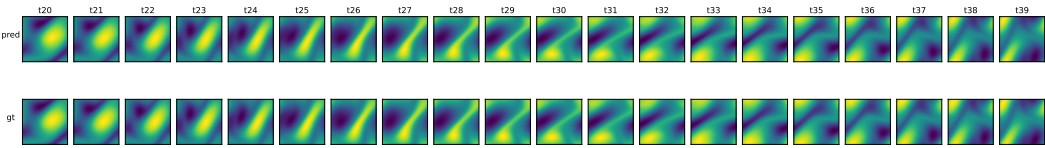

Figure 11: Test example rollout trajectory with AROMA on *Navier-Stokes* and *Out-t*. The first row is the prediction and the second row is the ground truth.

Figure 17, Figure 18, Figure 19. As evidenced in these figures, the perturbation of a token changes only locally the reconstructed field, which validates the spatial structure of our tokens. Notice in Figure 14 and in Figure 19 how these token affect the reconstruction in regions close to the boundary conditions. Our encoder-decoder has discovered from data and without explicit supervision that the solutions had periodic boundary conditions.

### D.3 LATENT SPACE DYNAMICS

For *Navier-Stokes*, we show how the mean (Figure 20) and standard deviation tokens (Figure 21) evolve over time for a given test trajectory. We show the predicted trajectory in the latent space in Figure 22.

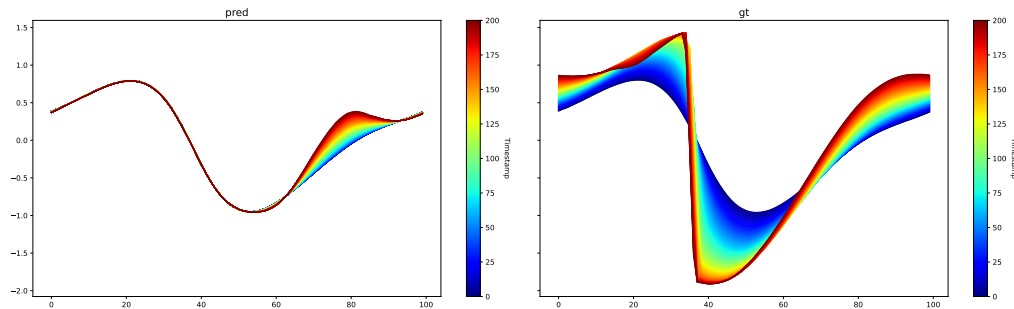

Figure 12: Perturbation analysis on *Burgers*. Token 0.

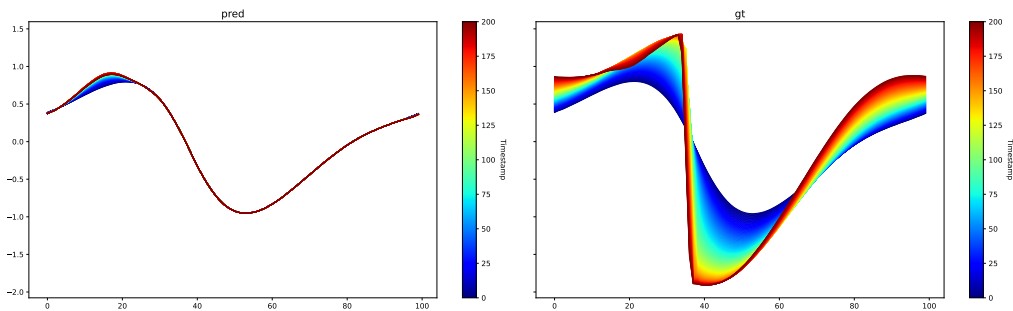

Figure 13: Perturbation analysis on *Burgers*. Token 1.

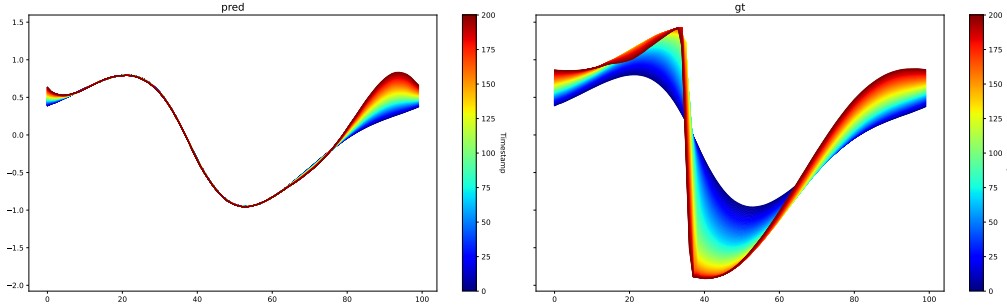

Figure 14: Perturbation analysis on *Burgers*. Token 2.

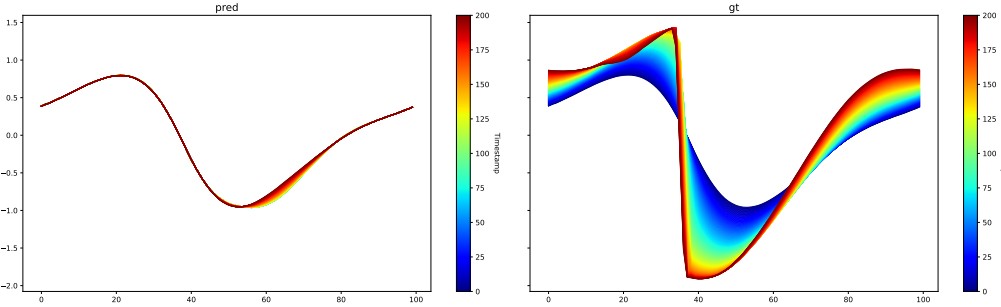

Figure 15: Perturbation analysis on *Burgers*. Token 3.

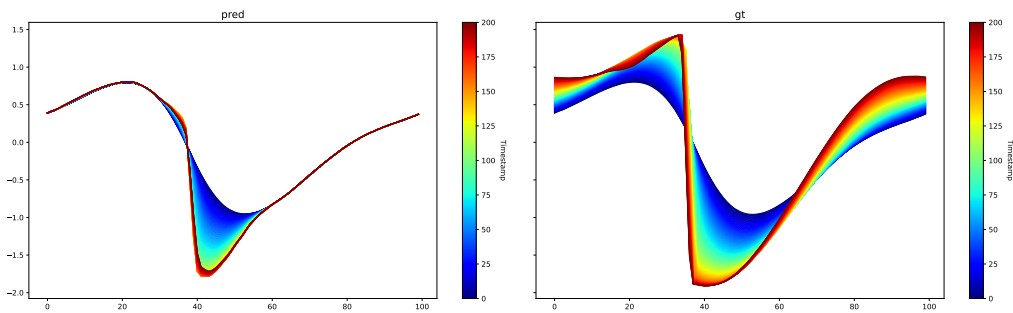

Figure 16: Perturbation analysis on *Burgers*. Token 5.

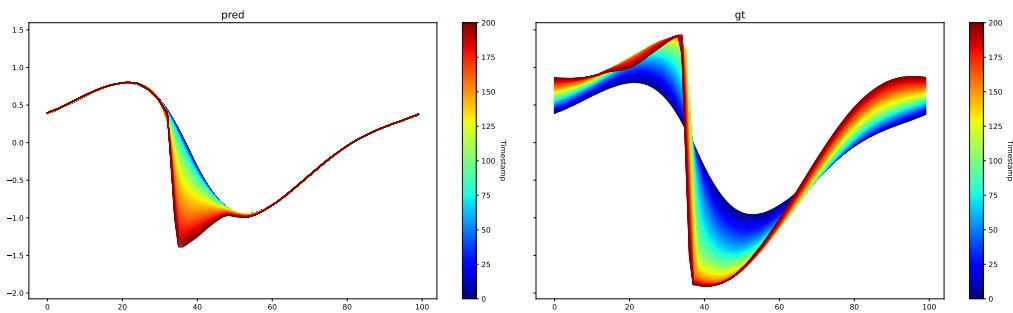

Figure 17: Perturbation analysis on *Burgers*. Token 6.

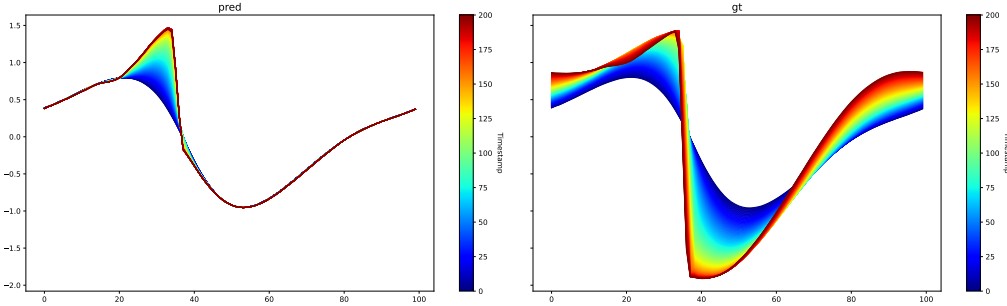

Figure 18: Perturbation analysis on *Burgers*. Token 7.

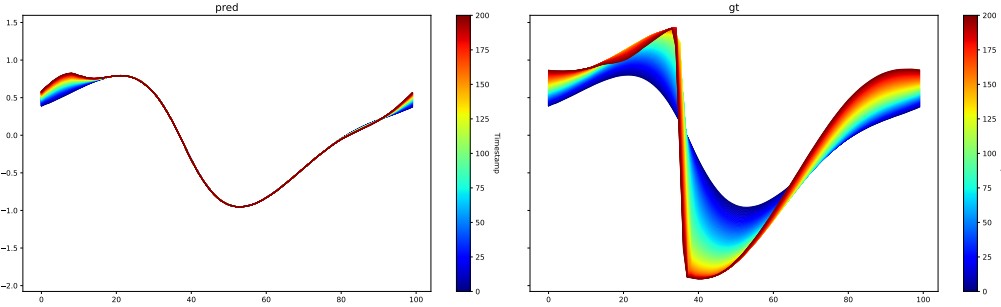

Figure 19: Perturbation analysis on *Burgers*. Token 8.

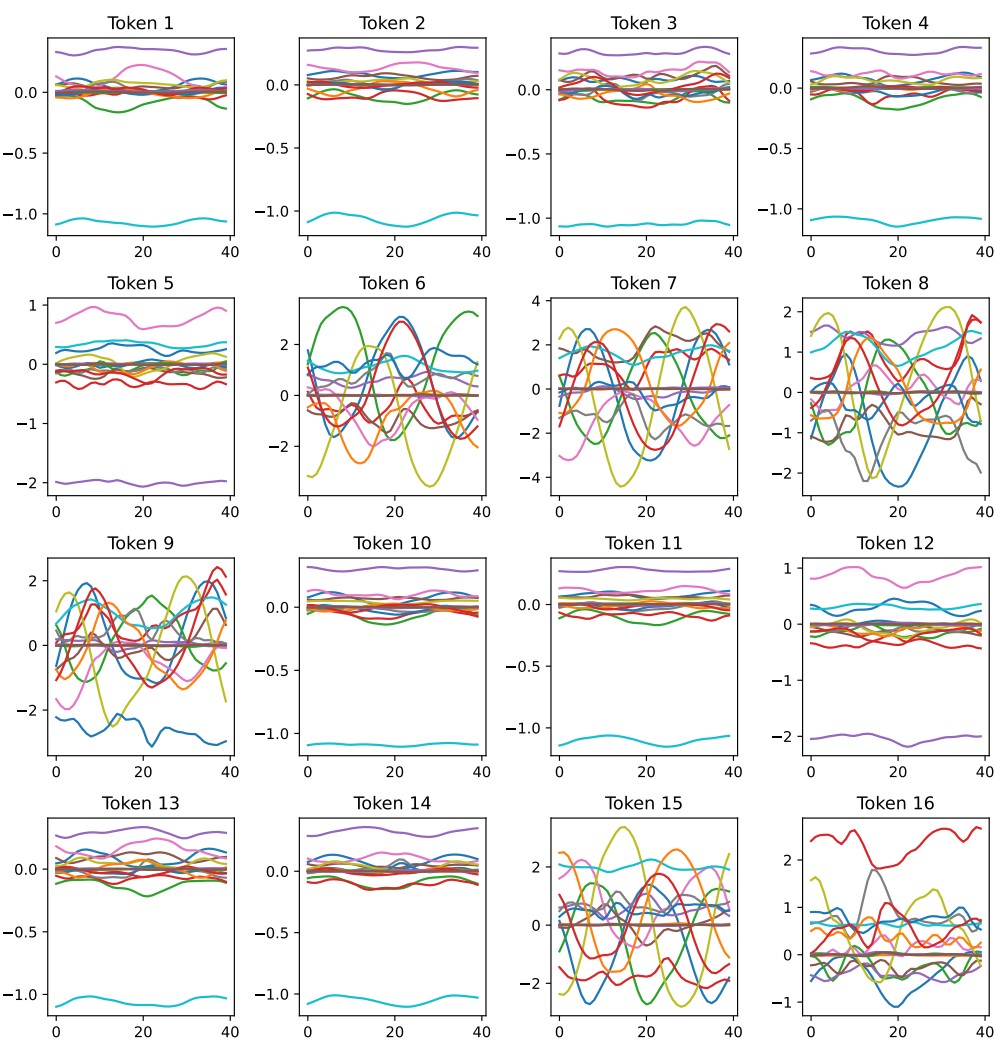

Figure 20: Latent space dynamics on *Navier-Stokes* - Mean tokens over time. Each color line is a different token channel.

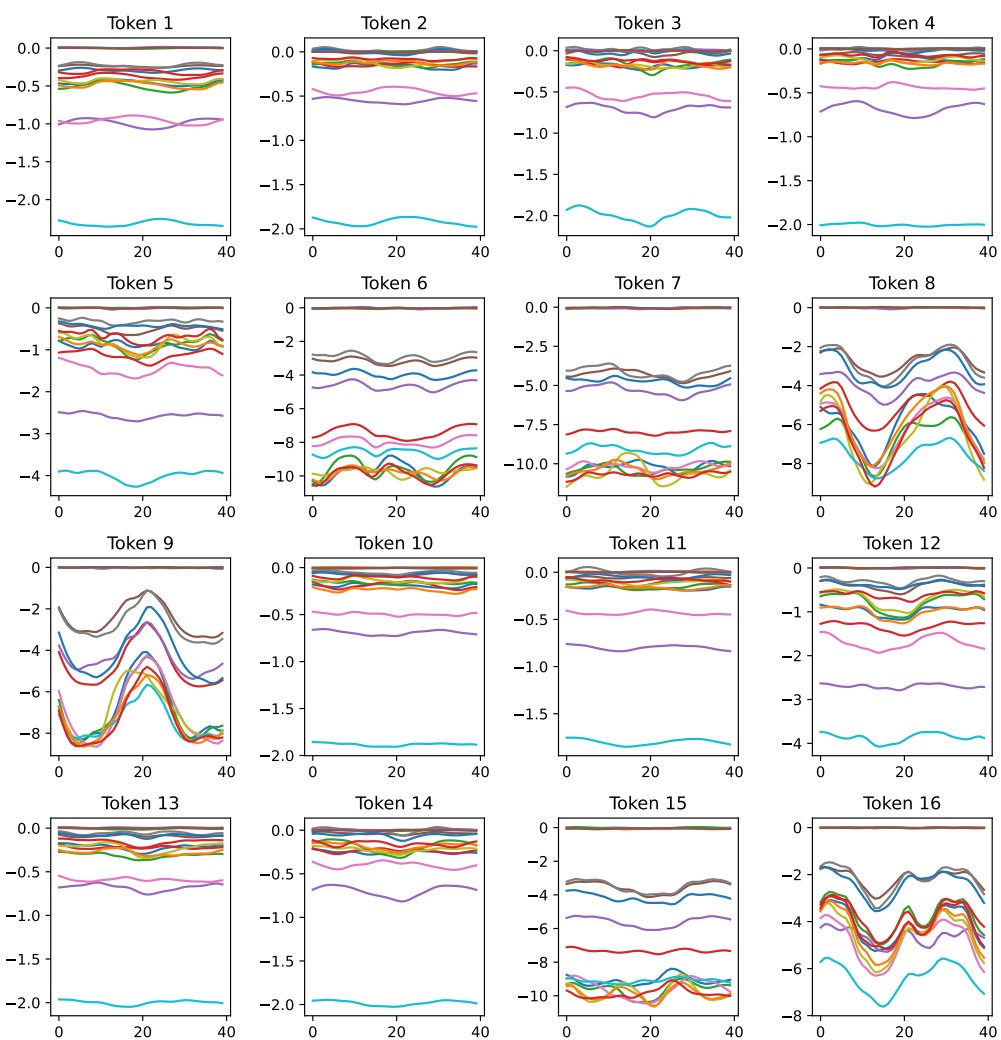

Figure 21: Latent space dynamics on *Navier-Stokes* - Logvar tokens over time. Each color line is a different token channel.

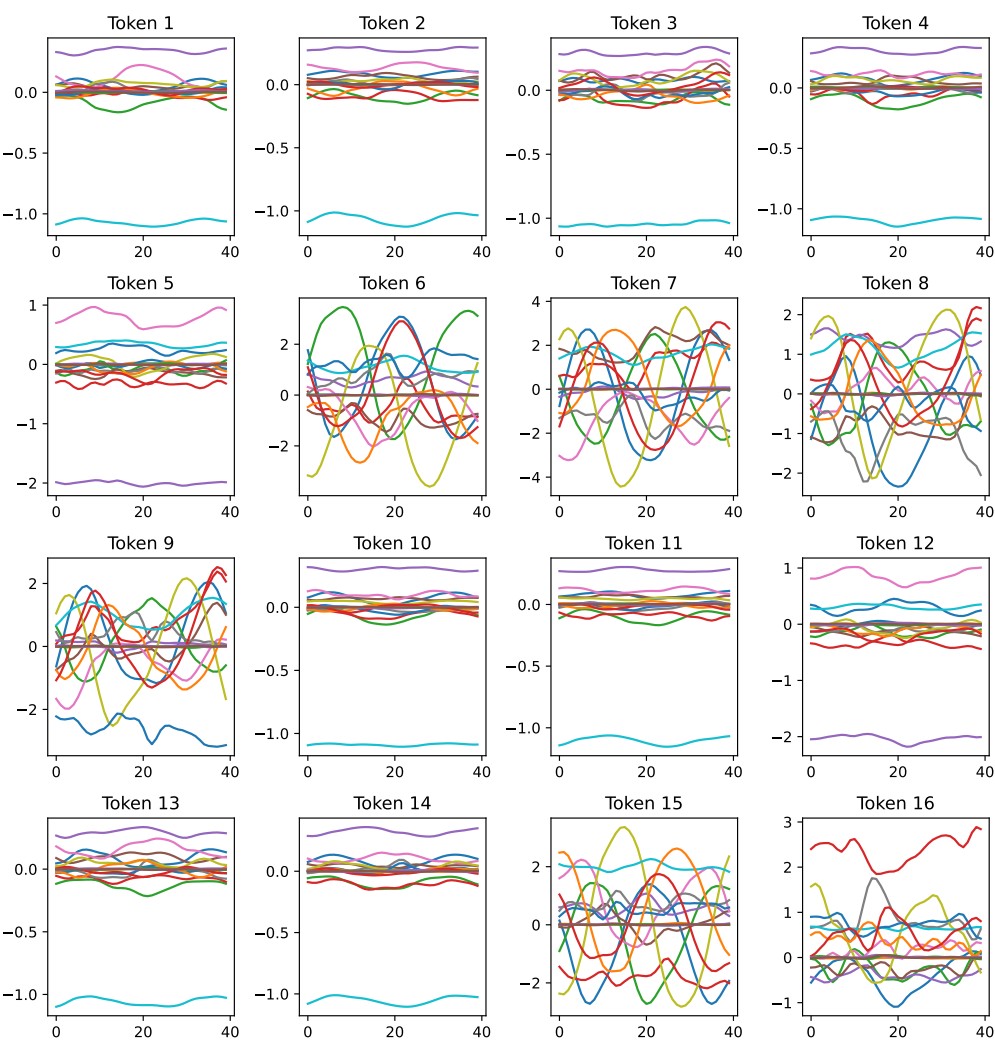

Figure 22: Latent space dynamics on *Navier-Stokes* - Predicted tokens over time. Each color line is a different token channel.

