# OpenReview forum: "Latent Diffusion Transformer with Local Neural Field as PDE Surrogate Model"
_ICLR.cc/2024/Workshop/AI4DiffEqtnsInSci — AI4DiffEqtnsInSci @ ICLR 2024 Poster_

### Official Review · Reviewer_SQ2z · 2024-02-26

**Rating:** 7
**Confidence:** 5

**Review:**

The premise of the paper is very interesting-- leveraging discretization-free INR representation without losing the benefits of a (somewhat) local representation.
Method-wise there's a lot going on in this paper-- a latent-space transformer, a diffusion model for next-step prediction and an INR decoder. This is good for a workshop, there's a lot of food for thought and things to discuss.

There isn't much validation of the papers' ideas so far, experiments are basically one comparison study. So currently we don't learn too much from the method choices-- I would definitely encourage the authors to keep exploring this space, drilling down on the individual components, and running ablations. E.g. what does setting this up as a diffusion model buy? Or maybe you get really good results for a diffusion transformer with a plain grid-based VAE decoder.
Focussing on one key element that works well, and really studying it could make an excellent full conference paper.

---

### Official Review · Reviewer_7rSp · 2024-02-27
**Advanced architecture for neural PDEs**

**Rating:** 4
**Confidence:** 5

**Review:**

The authors suggest using a combination of diffusion model and a cross-attention layer to learn latent variables in a neural PDE formulation. Despite a more complicated architecture that must have resulted in more GPU-hours than the baselines, the numerical results were not noticeably different. The authors should provide a comparison on training time of their model vs the baseline and further numerical justification for their idea.

---

### Meta-Review · Area_Chair_n2c8 · 2024-03-01

**Recommendation:** Accept (Poster)

**Metareview:**

This paper proposes to combine diffusion model and a cross-attention layer for dynamics modeling of complex systems. The concerns raised by the reviewer 7rSp are valid. I strongly command the author to include the training time of their model vs the baseline to present a more comprehensive picture.

---

### Decision · Program_Chairs · 2024-03-01

Accept (Poster)